# The Interplay of Autophagy and Tumor Microenvironment in Colorectal Cancer—Ways of Enhancing Immunotherapy Action

**DOI:** 10.3390/cancers11040533

**Published:** 2019-04-14

**Authors:** Evangelos Koustas, Panagiotis Sarantis, Georgia Kyriakopoulou, Athanasios G. Papavassiliou, Michalis V. Karamouzis

**Affiliations:** 1Molecular Oncology Unit, Department of Biological Chemistry, Medical School, National and Kapodistrian University of Athens, 11527 Athens, Greece; vang.koustas@gmail.com (E.K.); psarantis@bioacademy.gr (P.S.); gpkyriakopoulou@yahoo.com (G.K.); papavas@med.uoa.gr (A.G.P.); 2First Department of Internal Medicine, ‘Laiko’ General Hospital, Medical School, National and Kapodistrian University of Athens, 11527 Athens, Greece

**Keywords:** Autophagy, colorectal cancer, immunotherapy, tumor stroma, tumor microenvironment

## Abstract

Autophagy as a primary homeostatic and catabolic process is responsible for the degradation and recycling of proteins and cellular components. The mechanism of autophagy has a crucial role in several cellular functions and its dysregulation is associated with tumorigenesis, tumor–stroma interactions, and resistance to cancer therapy. A growing body of evidence suggests that autophagy is also a key regulator of the tumor microenvironment and cellular immune response in different types of cancer, including colorectal cancer (CRC). Furthermore, autophagy is responsible for initiating the immune response especially when it precedes cell death. However, the role of autophagy in CRC and the tumor microenvironment remains controversial. In this review, we identify the role of autophagy in tumor microenvironment regulation and the specific mechanism by which autophagy is implicated in immune responses during CRC tumorigenesis and the context of anticancer therapy.

## 1. Introduction

Colorectal cancer (CRC) is the third most frequently diagnosed malignancy and the second leading cause of cancer-related deaths in the U.S.A. and worldwide [1]. By 2030, the estimated global burden of CRC is expected to reach more than 2.2 million new cases and 1.1 million deaths [2]. Despite significant advances in standard of care therapies, the 5-year survival rate for patients diagnosed with metastatic CRC remains very poor, at approximately 12% [1]. Among others, autophagy is a major mechanism which is strongly associated with tumorigenesis in different types of cancer, including CRC. 

The mechanism of autophagy has been identified as a catabolic process with an essential role to digest proteins and dysfunctional cellular organelles [3]. Numerous steps related to autophagy include membrane trafficking vesicles, essential autophagy proteins, a double membrane organelle, which is called an autophagosome, and fusion with lysosomes to create the autophagolysosome. Autophagolysosome is a fundamental structure responsible for degrading the luminal content [4]. The role of autophagy is extended from cellular homeostasis to tumor development [5,6].

Many genes and proteins are crucial for the initiation and progression of autophagy. Genes, like *Beclin-1*, *LC3*, *ATG5*, and *ATG6*, have a crucial role for autophagy from normal function to CRC, where these genes have been reported with high expression. Furthermore, these autophagy gene-markers are associated with a more aggressive CRC phenotype [7].

Various morphological changes characterize the autophagy process. In the first step of autophagy, which is called initiation or nucleation, the phagophore, a double membrane structure, is formed through the activation of the class-III PI3K-Beclin-1 complex. Elongation is the next step in the autophagy process. This step is characterized by the arising of the phagophore from different double membrane organelles, such as the Endoplasmic Reticulum (ER), Golgi, and mitochondria. The phagophore starts to enclose the cytosolic cargos, leading to the formation of the autophagosome. The formation of the phagophore is highlighted by different Atgs, p62/ SQSTM1 (an adaptor protein responsible for the docking of specific cargoes), and the lipid-modification of LC3I to LC3II. The maturation step and the following fuse step include the autophagosome formation, which eventually fuses with lysosomes to form autolysosomes. Finally, during the degradation step, lysosomal/vacuolar hydrolases digest autolysosomal products and release them in the cytosol [4].

Over the last years, many studies have been conducted that support the dual role of autophagy in CRC. Autophagy appears to be responsible for maintaining the energy homeostasis in cells, which is required for several cellular functions, such as proliferation [8], angiogenesis, migration [9], and EMT (epithelial-mesenchymal transition) phenotype [10]. Autophagy is identified to be upregulated in a hypoxic region of already established tumors, where the energy demands are increased [11]. Moreover, cancer cells of high graded tumors appear to be addicted to autophagy to maintain their energy balance [12,13]. Numerous studies report the impact of autophagy in cancer patients’ response to chemotherapy. Increasing levels of autophagy are linked with inadequate response to chemotherapeutic drugs and dismal survival rates [14].

In different cancer types, such as CRC, a single-nucleotide polymorphism, in autophagy-related genes, like *ATG16L1*, is associated with a reduction of autophagy and a significant negative predictive value for patients’ survival with metastatic disease [15,16]. Besides, monoallelic deletion of other crucial autophagy genes, such as *Beclin-1*, which leads to autophagy reduction, has been identified in several diseases, such as cancer and Alzheimer’s [17,18,19]. Other studies highlight the positive impact of monoallelic deletion or total loss of other genes, such as *ATG5*, *ATG7*, and *ATG4C*, in cancer development [20]. In addition, *KRAS*, an essential oncogene in CRC development, is strongly associated with autophagy [21]. Cancer cells of *KRAS*-dependent tumors use autophagy in order to support the growth of cancer cells under stressful conditions in hypoxic regions of tumors [8]. All these studies highlight the dual role of autophagy as a tumor promoter or tumor suppressor mechanism. The accumulation of dysfunctional proteins and cellular organelles through the reduction of autophagy increases the risk for malignant transformation. Furthermore, low basic levels of autophagy are required for cell survival as was identified through experiments with a knockout of different autophagy genes, such as *ATG* genes, *Beclin-1*, or *AMBRA1* [22,23]. Autophagy is responsible for recycling cellular components and producing energy and pro-oncogenic factors [24]. Different stage of tumors, anti-cancer treatment, mutations in *ATGs*, and oncogenes are closely associated with autophagy and its controversial role in tumorigenesis. Further study is required in order to address the link between autophagy and hallmarks of cancer. 

Furthermore, the increasing levels of autophagy, in these regions, are strongly associated with the regulation of the immune response in the tumor microenvironment [11,25]. The microenvironment of different malignant tumors, including CRC, is characterized by numerous cell types (including immune, tumor, and other types of cells). All these stroma cell types utilize a different extent of autophagy. Therefore, focusing on autophagy and its role in the tumor microenvironment for the discovery of novel anti-cancer therapeutic targets should be further elucidated [11,26]. The role of autophagy in developing an immune response against tumor cells is far more complex. Therefore, autophagy may be a promising therapeutic target in combination with other anti-neoplastic drugs and immunotherapy in the context of this unique cellular composition of the tumor microenvironment.

## 2. The Major Players in the Tumor Microenvironment

For years, solid cancers were considered as a mass of homogenous cancer cells [27]. Cancer evolution and resistance to treatment is caused by tumor heterogeneity. Over the past decade, it has become increasingly clear that there is a wild diversity of cells with tangled and branching pedigrees in the same tumor. One section of a tumor might be dense with cells containing a particular oncogene mutation, whereas another section might have vastly different mutation backgrounds driving their growth [28]. Tumors should be perceived as separate tissues with a different and more complex cellular network with specialized or dedifferentiated malignant cell types, fibroblasts, tumor stem cells, immune, and endothelial cells. This complex network is characterized as a tumor stroma with unique potential for anticancer therapy [29].

### 2.1. The Heterogeneity of the Tumor Microenvironment

The vast majority of solid tumors are composed of not only malignant cells, but also of fibroblasts. It is widely accepted that tumorigenesis is a multistep process, the progression of which depends on a sequential accumulation of mutations within tissue cells. Moreover, tumor initiation is associated with the activation of different stromal, endothelial and mesenchymal cells, fibroblasts, and immunogenic cells [30,31]. It is well known that tumor heterogeneity is associated with the more aggressive phenotype and a lack of response against anti-cancer therapy in different types of cancer, including CRC [32].

### 2.2. The Role of T-Lymphocytes

The major effectors of the immune response against tumor cells are the cytotoxic CD8+ T-lymphocytes or T-cells (CTL). The abundance of T-cells is a decisive prognostic factor for the response of chemotherapy and immunotherapy in cancer patients especially at early stages of the disease, where patients have a strong effector T cell response and more frequently present a high Immunoscore [33,34]. CTLs are responsible for killing hostile cells, such as tumor cells [35]. Type 1 of T-helper cells (Th-1) regulates the activation of CTL and Th-2 initiates humoral immunity [36]. In many studies, the activation of the immune system and tumor-infiltrating lymphocytes (TILs) are used for the grading of the tumor and as a putative prognostic marker for CRC patients. The characterization is based on TILs, tumor invasion, spread to the lymph nodes, and the tumor staging system [33,35]. 

Many studies have identified that the activation of CTL is inhibited by the PD-L1/PD1 axis interaction in CRC tumors with the Mismatch repair deficiency/Microsatellite instability -high MMRd/MSI-H phenotype [37,38,39]. The clinical effectiveness of anti-PD1 monoclonal antibodies is beneficial for this subgroup of patients [40]. In contrast, with MSI-H CRC tumors, in almost all MSS CRC tumors, inhibition of the PD-L1/ PD1 axis has no significant clinical effect, thus underlining the complexity of this immunosuppressive mechanism [41].

A particular group of lymphocytes that are strongly associated with tumors is the regulatory T-cells [42]. The role of Tregs (regulatory T cells) is controversial because of the genetic and phenotypic differentiation of T-cells. The Treg-specific DNA hypo-methylated regions contribute to the stable expression of Treg function-associated key genes, including *Foxp3*. Accordingly, *FoxP3* robustly represses different genes, including *Il2*, contributing to Treg suppressive activity. In tumors, it is critical to deplete *FOXP3* high CD45RA_CD25 high effector Treg cells, which are firmly installed with the Treg-type hypo-methylation and are most suppressive [43]. The origin of Tregs can be either directly from the thymus (tTreg) or by peripheral differentiation (pTreg) of conventional T lymphocytes [44]. The majority of Tregs are characterized by a high expression of specific biomarkers such as FOXP3, IL-2 receptor alpha chain, CD25 IL-10, TGF-β, and IL-35. Also, proteins, like CTLA-4 (cytotoxic T-lymphocyte–associated antigen 4), PD-1 (programmed death 1), and GITR (the receptor of glucocorticoid-induced tumor necrosis factor), have been identified in the surface of Tregs [45,46,47]. It is well known that molecules, like IL-27 and IL-33, are stimulators of Tregs in CRC through TGF-β-mediated differentiation of Tregs [44].

The primary role of Tregs is to control inflammation and maintain peripheral tolerance in immune homeostasis. Furthermore, FOXP3+ Tregs are crucial in the inhibition of the cytotoxic effect of T-cells in many cancer types, including CRC [42]. The lack of FOXP3+ Tregs and the ratio of CD3+/FOXP3+ T cell may be a prognostic marker for clinical outcomes in patients with CRC [48].

### 2.3. The Role of Tumor-Associated Myeloid Cells

Different cell types, such as cancer-associated fibroblasts (CAFs) and tumor-associated macrophages (TAMs), in the tumor microenvironment, regulate tumor growth, invasion, and the metastatic phenotype of cancer cells [49,50]. Many studies support the hypothesis that bone marrow-derived cells (TANs, TAMs, and myeloid-derived suppressor cells or MDSCs) are closely associated with the progression of the tumor [50,51].

Two different sub-populations of TAMs, the anti-tumorigenic and pro-tumorigenic or M1 and M2 phenotype, respectively, with high plasticity, have already been identified [52,53]. The most common myeloid infiltrate in solid tumors is composed of myeloid-derived suppressor cells (MDSCs) and tumor-associated macrophages (TAMs). These cells promote tumor growth through their inherent immunosuppressive activity, neoangiogenesis, and mediation of epithelial-mesenchymal transition. Several small molecules are already used in order to inhibit the tumorigenic action of these cells [52]. It is well known that neutrophils regulate the tumor microenvironment through the production of several immunogenic, angiogenic, and inflammatory factors, such as matrix metalloproteinases (MMPs), Vascular endothelial growth factor (VEGF), neutrophil elastase, and hepatocyte growth factor [54,55,56]. The number of neutrophils in peripheral blood is already evaluated as a negative clinical progression marker in various malignant tumors, including CRC [57]. The two different types of neutrophils, N1 and N2 neutrophils, have been associated with tumor progression. N1 neutrophils reduce tumor immunosuppression through the production of several molecules, such as TNF-α, ROS (Reactive oxygen species), ICAM-1 (Intercellular Adhesion Molecule 1), and Fas. In contrast, N2 neutrophils, increase tumorigenicity through the production of MMP-9, VEGF, and several chemokines [58].

Myeloid-derived suppressor cells or MDSCs have an immunosuppressive ability that is triggered by inflammation. MDSCs are abundant in different tumors types with a critical role in tumor progression [56]. Tumors produce several chemokines, such as CCL2 and CCL5, which regulate the migration of MDSCs in tumors [59]. Several studies support the idea that tumors attract MDSCs in the tumor microenvironment. MDSCs suppress the anti-tumor activity of the immune system through the activation of different genes associated with arg1 (Arginase 1), fatty acid oxidation (FAO), and ROS [60]. Furthermore, MDSCs seem to inhibit both antigen-specific and nonspecific (CD3/CD28) proliferative responses in the tumor microenvironment in both ROS-dependent and independent ways. Also, MDSCs inhibit the stimulation of CD3/CD28 T-cells through the production of NO (Nitric Oxide) and Arg1 [61]. In the tumor microenvironment, MDSCs are converted into nonspecific suppressor cells through the up-regulation of iNOS (inducible nitric oxide synthase) and arginase I. These enzymes are known to be actively involved in T cell suppression in a way that does not require antigen-specific contact between MDSC and T cells to inhibit their function [62].

Several studies over the last years highlight the impact of autophagy in MDSCs’ survival in the tumor microenvironment. Glycolytic metabolism is strongly associated with the metabolism of MDSCs [63]. Glycolysis prevents the AMPK-ULK1, a key player in autophagy regulation, which increases the GM-CSF (granulocyte macrophage colony-stimulating factor) expression and supports the development of MDSCs in the tumor microenvironment [64]. Furthermore, MDSCs activate autophagy through phosphorylation of AMPK. The initiation of autophagy increases several anti-apoptotic factors, such as BCL-2 (B-cell lymphoma 2) and MCL-1 (Myeloid cell leukemia 1), which promotes multiple myeloma (MM) progression [65].

### 2.4. Cancer-Associated Fibroblasts (CAFs)

Cancer-associated fibroblasts or CAFs represent a heterogeneous group of cells. They are responsible for the remodeling of the extracellular matrix (ECM) and support the invasion and metastasis of cancer cells [66]. Different molecules, such as FAP (fibroblast activation protein) and alpha-smooth muscle actin (a-SMA), have been already used as markers of activated CAFs and other fibroblasts [67]. CRC transcriptome studies associate the presence of CAFs with poor outcomes of patients, thus underlining the clinical significance of CAFs as a prognostic marker. Furthermore, the differentiation of CAFs and induction of the fibrogenic phenotype is regulated by the signaling pathway of TGF-β, mechanical stress, and fibronectin [68,69,70].

### 2.5. Angiogenesis and Neo-Vascularization Process in Tumor Stroma

It is well known that the stroma of CRC is also the scaffold for the development of tumor-associated blood vessels. Mesenchymal cell type, such as fibroblasts and immune cells, are responsible for supplying the VEGF with tumors cells [71]. Other molecules, like MMPs and associated proteases, that are expressed by immunosuppressive myeloid cells (IMCs) and CAFs appear to be increased in the tumor microenvironment. These enzymes help neo-angiogenesis by altering the ECM and proteolytic activation of embedded angiogenic factors (FGF and VEGF) [72].

### 2.6. Other Immune Cell Types in the Tumor Microenvironment

Several studies identified many other immune cell types in the tumor microenvironment of CRC. Immune cell types that appear in CRC microenvironment, like neutrophils, mast cells, natural killer (NK) cells, or eosinophils, did not appear to have a significant role in the impact of the clinical progression of CRC patients [73,74].

## 3. The Role of Autophagy in Stroma Development, Inflammation, and the Immunity Response

It has been proven that autophagy affects the microenvironment of the tumor and vice versa. These microenvironmental factors include cytokines, hypoxia, and inflammation in the tumor environment [75]. In response to stress conditions in the tumor microenvironment, autophagy is activated to maintain and supply energy. Additionally, digestion of intracellular components prevents the accumulation of toxic cellular remnants.

Cancer cells coexist with their microenvironment and the role of autophagy in modulating their interactions with other cell types may be a target for the modulation of autophagy, as a potential anti-cancerous treatment [76]. Autophagy is also a key factor in the function of APCs and T-cells. Autophagy is implicated in the presentation of antigens in both MHC-I and MHC-II in Dendritic cells (DCs). Finally, autophagy contributes to the functional activity of immune cells by creating T-cell memory, depending on autophagy [77].

### 3.1. The Role of Inflammation in Colorectal Cancer Development

Chronic inflammation is a high-risk factor for cancer. Patients with inflammatory bowel disease (IBD), including Crohn’s disease (CD) and ulcerative colitis (UC), have a three-fold increased risk of developing CRC. This type of cancer is known as “colitis-associated colorectal cancer (CAC)” [78]. Activation of Toll-like receptor 4 (TLR4) promotes the development of colitis-associated cancer through activation of the Cox-2 and EGFR signaling pathway [79]. Cancer development is due to the non-neoplastic inflammatory epithelium. Mutations in essential genes (*c-src*, *p53*, *K-RAS*, *β-catenin*, and *APC*) are caused by inflammation as well as DNA damage, which then leads to CAC onset in patients with IBD. Moreover, inflammation triggers signaling pathways, such as STAT3 (Signal transducer and activator of transcription 3) and β-catenin, which causes proliferation and remodeling of epithelial cells and then promotes tumor growth [80]. The CAC microenvironment is a complex system of various types of cells, cytokines, and signaling molecules that play a significant role in tumorigenesis. Immune cells develop many individual functions in the CAC microenvironment. Macrophages promote CAC tumorigenesis and the development of reactive oxygen species (ROS), IL-5, and nitric oxide synthase (NOS) [80]. Tregs and Th17 cells have tumor-promoting activity during CAC [81,82] formation while CD8+ T cells serve a protective role against CAC oncogenesis [83]. TAMs and CADs regulate the production of cytokines, such as IL-6, IL-8, IL-10, and IFN-γ, in the tumor microenvironment. Cytokines are key molecules to the development of inflammation during tumor progression [84]. Several studies support that autophagy is triggered via inflammation. In addition, NLRP3 (NLR Family Pyrin Domain Containing 3) inflammasome (a mitochondrion that is damaged depending on the structure) is negatively regulated by autophagy with IL-1b and IL-18 production and subsequent inflammation response under control [25].

### 3.2. Hypoxia-Induced Autophagy in the Tumor Microenvironment

Many studies have shown that many types of tumors are found under hypoxic conditions [4,26]. Autophagy in a hypoxic environment in tumors depends on the duration and percentage of hypoxia. Under moderate and chronic hypoxia, hypoxia-induced factor-1 (HIF-1a) and PKC-JNK regulate autophagy [85]. Since hypoxia results in BNIP3 or REDD1 being dependent on autophagy, the question arises as to whether there is an association between BNIP3, HIF-1, and/or REDD1. Many published data support the notion that HIF-1α can up-regulate BNIP3 transcription. Enhanced BNIP3 then interferes with the Beclin1 and BCL2-forming complex and further suppresses Rheb-mTOR [86,87]. Hypoxia raises the levels of REDD1, which then separates the 14-3-3 proteins from the TSC2 complex and finally reduces mTOR [87]. Also, a stress sensor, Ataxia Telangiectasia Mutated (ATM), was verified as being involved in the REDD1-modulated mTOR signaling. Under the hypoxic environment, ATM (Ataxia Telangiectasia Mutated) (-/-) MEFs perform decreased expressions of HIF-1α and REDD1. Overall, it is suggested that hypoxia-induced ATM activation results in increased HIF-1α-BNIP3 and REDD1 to increase autophagy through the inhibition of mTOR [87,88].

### 3.3. The Cross-Talk between Autophagy and Antigen Presenting Cells

Activation of the anticancer T-cell is induced by identifying the antigenic tumor peptides present on the cell surface of professional APCs, like DCs. However, autophagy through DCs and macrophages affects the surface expression of the MHC-I and peptide complex. For example, the expression of MHC-I in embryo mice DCs and macrophages was upregulated during inhibition of autophagy using chemical inhibitors or downregulation of the main autophagy genes [89,90]. This adjustment was attributed to the slower internalization of classical MHC class I molecules, leading to increased CD8+ T cell stimulation [90]. Hence, in the absence of autophagy, MHC-I molecules appear more consistently expressed and less degenerated [91]. Equally, DCs from mice lacking VPS34 (vacuolar protein sorting-associated protein 34) expressed more MHC-I on the cell surface as well as MHC-II [92]. In contrast, surface expression of MHC-II in macrophages was downregulated when inhibiting autophagy using 3-Methyladenine (3-MA) [91]. Autophagy is associated with the cross-presentation of antigens in DCs. Cross-presentation is a process that permits the loading of MHC-I into DCs with extracellular antigens, which is essential for activating, for example, CTL responses in melanoma [91,93,94,95]. The cross-presentation capability of bone marrow-derived dendritic cells (DCs) is characterized by increased levels of autophagy [90,96].

Antigen presentation in MHC-II was similarly altered in the inhibition of autophagy with reduced DC treatment mediated by an immunodominant mycobacterial peptide with the reduced presentation of vaccinia virus Ankara antigens and herpes simplex virus (HSV) antigens [97,98]. Accordingly, antigen-specific T-cell responses were down-regulated. Thus, inhibition of autophagy modified the peptide pool presented in MHC and reducing the presentation of immunodominant epitopes [99]. Although, inhibition of autophagy up-regulates surface expression of MHC-I, it also changes the group of immunogenic peptides presented on MHC. Thus, the effect on surface expression of MHC-I and II is less well-confirmed, which has been best determined in the context of the so-called cross-presentation in DCs [93,100,101]. As it was mentioned before, increased levels of autophagy characterize the cross-presentation capability of DCs compared with DCs that do not cross-present antigens, and the autoimmune inhibition that reduces the cross-presentation of MHC-I mediated MHC-I [102,103]. Inhibition of autophagy modified the presentation of the different peptides in MHC and appeared to change the pool of immunodominant epitopes of these peptides. Further mechanistic studies are needed to define how autophagy serves as a target for MHC class I cross-presentation. The central role of autophagy in antigen-presenting cells (APCs) is presented in Figure 1.

In general, peptides are cleaved and digest from proteins through proteasome in the endogenous pathway. In the exogenous and cross-presentation pathway, the endocytotic peptides are closely associated with autophagy. Endosomes fuse with the autolysosomes in order to digest the peptides and the neo-antigens are loaded onto MHC I and II in the endoplasmic reticulum (ER). 

In the already developed tumor microenvironment, M2-phenotype tumor-associated macrophages (TAMs) promote angiogenesis, growth, and metastasis of tumor and cancer cells [104]. However, different studies support that M1 macrophages inhibit tumor growth [58]. The latest reports have shown that autophagy plays a crucial role in the production and polarization of macrophages. Deficiency of TLR2 strongly inhibits autophagy and leads to the biosynthesis of the M2 macrophage, which in turn promotes oncogenesis [58,105]. Moreover, the initiation of autophagy in TAM can increase the radiosensitivity of CRC, inhibit proliferation, and trigger apoptotic cell death [106].

Thus, autophagy in TAM can play a crucial role in cancer suppression. Also, the role of other native immune cells, such as NK cells and neutrophils, plays a vital role in the tumorigenesis of CRC. For example, tumor-associated neutrophils (TANs) facilitate the onset and development of CAC and increase autophagy in neutrophils, which are associated with increased migration of cancer cells [91]. Several in vivo studies suggest that inhibition of autophagy in tumor cells reduces the development of tumors by facilitating the removal of cancer cells via NK cells [107]. Analogous results have also been observed in other types of cancers, such as renal cell carcinoma and melanoma [81].

### 3.4. Autophagy—A Key Regulator of T-Cell Activation

The adaptive immune system includes the identification of pathogen or tumor proteins and their presence in MHC molecules by antigen-presenting cells (APCs). For this aim, MHC class I molecules are recognized by T cell receptors (TCRs) in CD8+ T cells. Subsequently, MHC class II molecules are recognized by TCRs in CD4+ T cells [90,91,92]. T cells are activated and differentiated into various types of effector T cells, including Tregs, Th cells, and cytotoxic T cells. Tregs produce anti-inflammatory cytokines, like IL-10 and TGF-β. Also, Th-cells can produce pro-inflammatory cytokines, such as IL-2, IL-5, IL-13, and IL-17A, and interferon gamma (INF-γ). Cytotoxic T cells cause the apoptosis of infected or malignant cells with the release of perforin and granzymes [81,108].

It has been reported that autophagy enhances the adaptive immune response by facilitating APC recognition and preserving the function, survival, and homeostasis of T cells among others [77]. T cell homeostasis involves the clearance of T cells deficient in autophagy [109]. For example, the loss of VPS34 accumulates ROS, which causes an increase in pre-apoptotic protein expression and robust apoptosis of these T cells [110]. Also, depletion of VPS34 also prevents the normal operation of Tregs. Moreover, the deletion of ATG5 and Beclin 1 results in inefficient proliferation and disordered function of CD8+ and CD4+ T cells, respectively, following TCR stimulation [111,112]. On the contrary, autophagy contributes to the maintenance of the survival and function of T cell lymphocytes CD8+ [113].

## 4. The Current State of Immunotherapy in CRC Patients

The treatment for CRC patients with early-stage disease is surgical removal of tumors. Chemotherapy usually follows the surgery for more advanced disease [114]. Recently, it has been shown that immunotherapy amplifies the immune responses against tumors and it has already been used for patients with solid tumors [115].

In the last few years, many immunomodulating agents have been developed that show significant efficacy. The FDA (Food and Drug Administration) has already approved immune checkpoint inhibitors, such as ipilimumab (an anti-CTLA-4 MoAbs), nivolumab, and pembrolizumab (anti-PD-1 MoAbs) or atezolizumab, avelumab, and durvalumab (anti-PD-L1 MoAbs) for different types of cancer, like melanoma, lung cancer, and renal cell carcinoma. They have recently shown promising activity as a treatment for CRC, although efficacy is reserved for a specific subset of patients [116,117].

It is well known that PD-L1, on tumor and stromal cells, suppresses the antitumor activity of the immune system through stabilization of TNF-α [118]. Furthermore, the PD-1/PD-L1 axis regulates inhibition of the immune response and leads T-cells to exhaustion and apoptotic cell death [119,120]. Wang et al. have shown that metastatic colorectal cancer (mCRC) has higher levels of PD-L1 [121]. Furthermore, dysregulation of signaling pathways, like PI3K-AKT, or chromosomal amplification of the 9q24.1 locus regulates the expression of PD-L1 and PD-L2 in different types of gastrointestinal cancers [120,122].

It is well known that the MSI phenotype in CRC varies according to the stage of the disease. CRC patients with mismatch repair (MMR) deficiency (15% to 20% of stage II/III CRCs) have a better prognosis. Metastatic CRC with deficient MMR represent around 5% and is associated with a poor prognosis [123]. Predictive biomarkers, like MMR and microsatellite status, a mutation in proto-oncogenes, and the expression of PD-L1 have already been used to classify patients in whom immunotherapy might be more beneficial [116,124]. Unfortunately, the percentage of patients with gastrointestinal cancer who will acquire durable clinical responses remains limited. The response rate for CRC patients with mismatch repair deficiency is less than 50% [125] and less than 30% for gastroesophageal cancer [125,126].

In many types of cancer, immunotherapy has been proven as a prominent therapeutic approach. Moreover, in the last few years, significant advances have also been made in CRC. An anti-CTLA-4 monoclonal antibody (tremelimumab) has proven useful for CRC patients, obtaining one 6-month strong response [127]. In a phase II trial, three groups of patients were formed according to their microsatellite status—MSI-H, non-MSI-H, and MSS CRC—in order to test the clinical activity of anti-PD1 MoAb, Pembrolizumab. The immune-related objective response rate (ORR) and immune-related 6-month PFS progression-free survival (PFS) rate were 40% and 78%, respectively, for mismatch repair–deficient (dMMR) colorectal cancers and 0% and 11% for mismatch repair-proficient colorectal cancers patients. The KEYNOTE-177 phase III trial evaluated the above results in patients with dMMR mCRC after treatment with Pembrolizumab versus standard therapy. In Checkmate 142, treatment with Nivolumab alone or in combination with Ipilimumab was tested in metastatic CRC patients according to the microsatellite status. In the update published on Lancet, 31% of CRC patients who were treated with Nivolumab had an objective response, with a disease control rate of 69% for 12 weeks or longer [123]. The combinatorial treatment of Nivolumab and Ipilimumab showed a 55% ORR, while the disease control rate for 12 weeks or longer was 80% [128,129].

The first anti-PD-L1 monoclonal antibody with FDA approval is atezolizumab. This is a fully humanized antibody which targets explicitly PD-L1. It is currently approved for patients with metastatic NSCLC and metastatic urothelial carcinoma with disease progression after treatment of platinum-based chemotherapy [130,131]. Atezolizumab shows response rates higher for patients with PD-L1 positive tumors [132,133]. A similar antibody is durvalumab. The safety and tolerability of durvalumab alone or in combination with tremelimumab have already been tested in a phase I trial for patients with CRC. Promising results have been presented in patients with PD-L1-expressing tumors with microsatellite instability [120,133,134]. These kinds of tumors are characterized by a higher number of infiltrated immune cells. 

Furthermore, anti-PD-L1 therapy is more efficient in combination because of the differential expression of PD-1 and PD-L1 in the tumor microenvironment. On the other hand, several types of cancers, such as melanoma and breast cancer, are characterized by PD-L1 expression in both tumors and infiltrating immune cells [120]. The other, a less studied ligand of PD-1 is PD-L2. PD-L2 has been identified to be expressed in macrophages, B-cells, and dendritic cells [124,135]. In CRC, the expression of PD-L2 is approximately 40% and it is regulated by glycosylation and IFNγ [136]. Further, ongoing studies are evaluating the combinations of PD-1, PD-L1, and/or CTLA-4 monoclonal antibodies with other chemotherapeutic molecules, which will re-activate the immune system against CRC tumors (Table 1).

Several studies associate the expression of PD-L1 with PD-L2 and with the geographical association of different types of immune cells. The protein levels of PD-L1 and PD-L2 are associated with the response of anti-PD1 MoAbs. Thus, PD-L2 may be a promising target in immunotherapeutic schemes for CRC [137,138]. It is well known that increasing levels of CD73 block the activation of lymphocytes via increasing adenosine levels. Thus, inhibition of CD73 enhances the therapeutic effect of anti-PD1 and anti-CTL4 monoclonal antibodies [139]. Furthermore, several studies, have explored the relationship between the inhibition of PD-1/CTLA-4 and the increasing levels of CD8+ and CD4+ T cells and cytokines, Tregs inhibition, and other molecules essential for T-cell function [120,140].

## 5. Targeting Autophagy—A Promising Anti-Cancer Strategy

### 5.1. The Main Autophagy Inhibitors in Cancer Therapy

Different studies in the last years support the concept of the protective role of autophagy against a different type of cancer therapy, like radiotherapy, chemotherapy, and immunotherapy [141]. The crucial role of autophagy is to regulate the energy and metabolic balance of cancer cells [17] and through the impairment of cell death [142]. Years of efforts have led to the development of molecules that inhibit autophagy. Because of the crucial role of autophagy in cancer cell initiation and progression, the inhibition of autophagy has been shown to be beneficial in anticancer treatment.

Chloroquine (CQ) and its derivative, hydroxychloroquine (HCQ), is one of the most well-known inhibitors that target the fusion of the autophagosome with a lysosome. Over the last years, different clinical trials have attempted to evaluate the clinical significance of autophagy inhibition with CQ or HCQ in several types of cancers [76]. Unfortunately, these clinical trials failed to provide clinically significant benefits because of a lack of consistent inhibition of autophagy with CQ and its derivative, HCQ [143]. However, the combination of autophagy inhibition with other agents provides some encouraging results [76,144]. The combination of HCQ with chemotherapy, like gemcitabine, in pancreatic adenocarcinoma reduced the level of tumor marker 19-9 around 60% [145]. Furthermore, inhibition of autophagy may also have benefits in immunotherapy. The combination of CQ with IL-2 has proven effective with limited toxicity in a preclinical murine hepatic metastasis model. Moreover, this combinatorial scheme increases long term survival and the proliferation and infiltration of immune cells in the liver and spleen [141].

The clinical response of CQ and HCQ appears to vary widely. CQ and its derivative, CHQ, are not specific inhibitors of autophagy [141] and this appears to affect the bioavailability of other drugs by altering the tumor pH [143,146]. Also, the lack of a specific biomarker, which evaluates the inhibition of autophagy, add to the difficulties of these autophagy inhibitors to provide clinically significant results. New, more specific autophagy inhibitors may provide benefits for patients [76,141].

A more potent autophagy inhibitor is Lys05, a dimeric for of Chloroquine. Lys05 alters the acidification of the lysosomes and causes impairment of lysosomal enzymes. It can be used in lower doses. Thus, it is more tolerated and associated with stronger anti-tumor activity [147]. Another autophagy inhibitor is SAR405. SAR405 is a specific kinase inhibitor of Vps18 and Vps34. Vps34 and Beclin-1 regulate the initiation of the autophagy process. Inhibition of Vps34 leads to dysfunctional lysosome and vesicle trafficking activity [148]. Several studies support that inhibition of Beclin-1 reduces tumor growth and enhances anti-tumor NK cell activity. Decreasing levels of Beclin-1 leads tumor cells to overexpress CCL5 cytokine, which regulates the trafficking of NK cells to the tumors [141]. SBI-0206965 is a highly selective, small molecule inhibitor for ULK1 (Unc-51 like kinase-1). This molecule inhibits autophagy through the reduction of ULK1-mediated phosphorylation events in cells. In vivo experiments support the antitumor activity of SBI-0206965 via inhibition of autophagy in different types of cancer [149]. Several other drugs, such as verteporfin, clomipramine, and desmethylclomipramine (DCMI), have been FDA-approved for use in therapy. All these agents alter the acidification of lysosomes or block autophagosome-lysosome fusion [150]. Specifically, autophagy inhibition through DCMI enhances the efficacy of doxorybicin in in vitro studies [151]. Another potent autophagy inhibitor is spautin-1. The mechanism by which spautin-1 inhibits autophagy has already been identified. It inhibits two ubiquitin-specific peptidases, USP10 and USP13, which regulate the deubiquitination of Beclin-1 in Vps34 complexes. Thus, autophagy initiation is inhibited [152]. Due to the strong association of autophagy with the tumor microenvironment and the immune response against tumors, autophagy inhibition may have a negative effect on the adaptive antitumor immunity against tumors. Starobinets et al. (2016) identified that adaptive antitumor immunity is not adversely associated with autophagy inhibition in breast and melanoma cancer models. Thus, autophagy inhibitors can be safely combined with other chemotherapeutic drugs, such as anthracyclines, and still trigger a productive antitumor T cell response against tumors [153].

### 5.2. Activators of Autophagy for Cancer Therapy

The current review attempts to extensively analyze the role of autophagy in the development of the tumor microenvironment and anti-cancer immunotherapy. In many cases, it is well understood that autophagy has a crucial role in the anti-tumor immune response in CRC. Autophagy not only regulates the antigen presentation in MHC I and II, but it has also been associated with apoptotic cell death in some cases. Due to the multifaceted role of autophagy in cancer, several molecules that induce autophagy have been developed in order to have benefits in anti-cancer therapy.

The most well-known autophagy activators are rapamycin and rapalogs (everolimus, temsirolimus, and deforolimus—analogs of rapamycin). They are inhibitors of mTOR and mTORC1, respectively, and consequently activate autophagy [154]. In endometrial cancer cells, everolimus has been identified as a suppressor of proliferation, especially when it is combined with paclitaxel [155]. Rapamycin was reported to enhance radiation therapy in A549 lung cancer cells through the induction of autophagy and delaying of DNA damage repair [156]. Rapamycin and rapalogs are putative therapeutic molecules that act through autophagy induction, especially when combined with other anti-neoplastic drugs. The clinical application of autophagy activators requires further investigation [155].

Another compound which reduces cell proliferation through the induction of autophagy is metformin. Inhibition of autophagy with specific autophagy inhibitors or knockdown of Beclin-1 reversed the cytotoxic effects of metformin. Furthermore, metformin was identified to increase TNF-related apoptosis-inducing ligand (TRAIL)-dependent apoptosis in lung adenocarcinoma cells through the induction of autophagy machinery [152]. In a BRCA1-deficient mammary tumor model, the combination of metformin with spautin-1 sensitizes BRCA1-deficient breast tumors to mitochondrial disruptors. It is well known that these two agents target different aspects of mitochondrial function and thus it may partially explain the contradictory observation of an autophagy inhibitor (spautin-1) with an autophagy inducer (metformin) in the reduction of cell viability [157].

Obatoclax, a molecule that specifically targets the Bcl-2 family, has been identified as an anti-cancer agent against hematologic malignancies [158]. The main anticancer mechanism of Obatoclax is strongly associated with autophagy induction. Furthermore, Obatoclax stimulates the assembly of necrosomes in the membranes of autophagosomes and consequently induces necroptosis [154]. Several studies have established natural alkaloids, such as isoliensinine, cepharanthine, and liensinine, as inducers of autophagy in cancer [159]. Alkaloids regulate autophagy through phosphorylation of AMPK and inhibition of mTOR. These kinds of alkaloids have been reported to induce apoptotic cell death in apoptosis-resistant MEFs [154].

Herein, we provide two summarized tables about small molecules that inhibit or activate autophagy. Regulation of autophagy is already used in research to develop new chemotherapeutic strategies with immunotherapy for different types of cancer, including CRC (Table 2 and Table 3).

## 6. Conclusions

In the last decade, autophagy has been strongly associated with tumorigenesis in colorectal cancer. The dual role of autophagy as survival and a pro-death mechanism has become a field of research in order to develop more effective therapeutic schemes against cancer. In established tumors, autophagy has a vital role as a survival mechanism, especially in the hypoxic regions of tumors. It is well known that tumors are characterized by a highly heterogeneous population of cancer, mesenchymal, immune, and stromal cells in a complex structure, which is identified as the tumor microenvironment. A growing body of evidence supports the hypothesis that autophagy regulates not only the metabolic function of cancer cells, but also other types of cells in the tumor microenvironment. Autophagy has a crucial role as a regulator of immune responses by sustaining the activation, homeostasis, and biological functions of different immune cells, such as T-cells, macrophages, and antigen presenting cells. Moreover, the impact of autophagy on tumor cells has also been observed in the active participation in intracellular and extracellular antigen processing for MHC-I and/or MHC-II presentation. Besides, autophagy has also been reported to associate with the cross-presentation of neo-antigens for MHC-I presentation and the internalization process. Several studies support autophagy as a potential target to strengthen or attenuate the effects of immunotherapy against different types of cancer, including CRC. In the future, efforts should be focused on how to regulate autophagy in the tumor microenvironment in order to strengthen the response of the immune system and overcome anti-tumor immune resistance in immunotherapy for colorectal cancer.

## Figures and Tables

**Figure 1 cancers-11-00533-f001:**
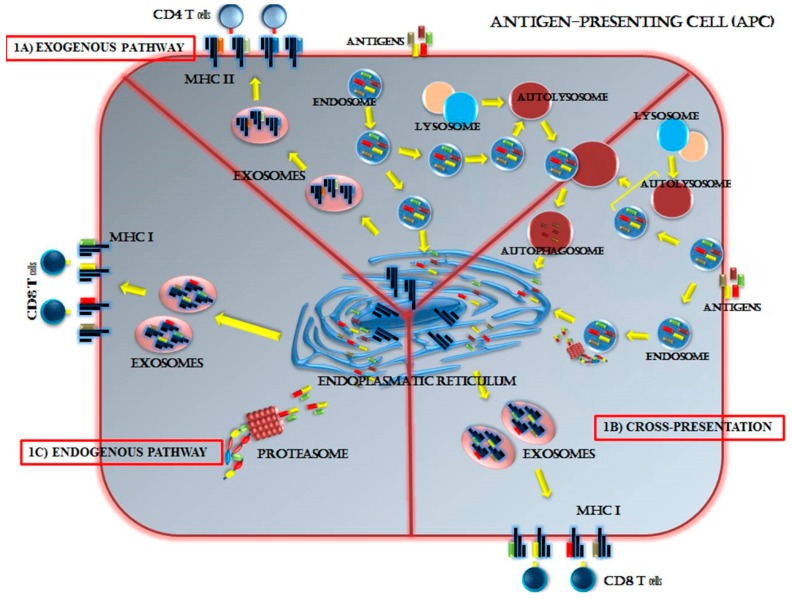
The role of autophagy in the presentation of immunogenic peptides in antigen-presenting cells (APCs). Autophagy has a vital role in the degradation of proteins in order for APCs to use them as antigenic peptides on Major Histocompatibility Complex (MHC)-I and II. Three distinct pathways of antigen processing by the APC have been identified: Exogenous (1A), cross-presentation (1B), and endogenous (1C) pathway. In the exogenous pathway, different antigens and peptides are produced outside the APC and placed on the MHC class II for recognition by CD4+ T cells. The exogenous pathway occurs in macrophages, dendritic cells, and B cells. The endogenous pathway loads cell-produced antigenic peptides onto MHC class I for recognition by CD8+ T cells. The endogenous pathway is responsible for immune recognition of peptides from the virus or self-digested peptides. The endogenous pathway characterizes many cell types, not just APCs, allowing for sensing of viral infection in all cell types. In the cross-presentation pathway, different peptides, from endocytosis and the autophagy degradation pathway, are loaded on MHC class I for recognition by CD8+ T cells. The peptides originate from the surrounding cell environment of tumor apoptotic bodies. This pathway targets virus-infected cells other than APCs and the tumor. The cross-presentation pathway is identified as the most efficient in dendritic cells.

**Table 1 cancers-11-00533-t001:** Clinical studies with immunotherapy for patients with Please define this term if appropriate.

Number of Study	Immune Target	Agent/Compound	Phase of Study
NCT01876511	PD-1	Pembrolizumab	II
NCT02981524	PD-1	Cyclophosphamide followed by Pembrolizumab	II
NCT03657641	PD-1	Pembrolizumab + Vicriviroc	I/II
NCT03631407	PD-1	Pembrolizumab + Regorafenib	II
NCT03475004	PD-1	Pembrolizumab, Bevacizumab, and Binimetinib	II
NCT03658772	PD-1	Pembrolizumab + grapiprant	I
NCT03519412	PD-1	Pembrolizumab + temozolomide	II
NCT02713373	PD-1	Pembrolizumab + cetuximab	I/II
NCT02375672	PD-1	Pembrolizumab + FOLFOX	II
NCT03332498	PD-1	Pembrolizumab + Ibrutinib	I/II
NCT02851004	PD-1	Pembrolizumab + SBRT	I/II
NCT02837263	PD-1	Pembrolizumab + BBI609	I
NCT02992912	PD-1	Atezolizumab + stereotactic ablative radiotherapy	II
NCT03712943	PD-1	Nivolumab + Regorafenib	I
NCT03711058	PD-1	Nivolumab + Copanlisib	I/II
NCT03414983	PD-1	Nivolumab, Oxaliplatin, Leucovorin, Fluorouracil, Bevacizumab	II/III
NCT02860546	PD-1	Nivolumab + TAS-102	II
NCT03026140	PD-1 and CTLA-4	Nivolumab + Ipilimumab +/− celecoxib	I/III
NCT03693846	PD-1 and CTLA-4	Nivolumab + Ipilimumab	II
NCT03104439	PD-1 and CTLA-4	Nivolumab + Ipilimumab + radiotherapy	II
NCT03377361	PD-1 and CTLA-4	Nivolumab + Ipilimumab + Trametinib	I/II
NCT03832621	PD-1 and CTLA-4	Nivolumab, Ipilimumab, Temozolomide	II
NCT02327078	PD-1 and IDO	Nivolumab + Epacadostat	VII
NCT02983578	PD-L1	AZD9150 + MEDI4736	II
NCT02982694	PD-L1	Atezolizumab + Bevacizumab	II
NCT02777710	PD-L1	Durvalumab + Pexidartinib	I
NCT03827044	PD-L1	Avelumab	III
NCT02669914	PD-L1	Durvalumab	II
NCT02754856	PD-L1 and CTLA-4	MEDI4736 + Tremelimumab	I
NCT03202758	PD-L1 and CTLA-4	Durvalumab, Tremelimumab, and FOLFOX	I/II

NCT, national clinical trial; PD-1, programmed cell death-1; PD-1, programmed cell death-1 ligand; CTLA-4, cytotoxic T-lymphocyte-associated protein 4; IDO, indoleamine-pyrrole 2,3-dioxygenase.

**Table 2 cancers-11-00533-t002:** Commonly used molecules inhibiting autophagy. Small molecules that have been identified as inhibitors of autophagy and the main mechanism of action.

Compound	Autophagy Inhibitors
	Mechanism of Action
Bafilomycin A1	Inhibitor of v-ATPase, inhibition of lysosomal acidification
Concanamycin A	Inhibitor of v-ATPase, inhibition of lysosomal acidification
Azithromycin	Inhibitor of v-ATPase, inhibition of lysosomal acidification
3-Methyladenine (3-MA)	Inhibitor of class III PI3K
Chloroquine (CQ)	Neutralizes the acidic pH of intracellular vesicles
Hydroxy-chloroquine (HCQ)	CQ derivative-Neutralizes the acidic pH of intracellular vesicles
Lys05	CQ derivative-alter the acidification of the lysosomes
SAR405	Kinase inhibitor of Vps18 and Vps34
SBI-0206965	Inhibitor of ULK1
Verteporfin	Inhibit acidification of lysosomes
Clomipramine	Inhibit acidification of lysosomes
desmethylclomipramine (DCMI)	Inhibit Autophagosome-Lysosome fusion
Paclitaxel	Microtubule stabilizer- inhibits phosphorylation of VPS34 at T159
SAHA	Interact in autophagosome-lysosome fusion
Monensin	Inhibit autophagosome-lysosome fusion
Sputin-1	Inhibits the activity of ubiquitin-specific peptidases, USP10 and USP13
SP600125	Inhibition of JNK—reduction of Beclin-1
U0126	Inhibitor of MEK1 and MEK2
Wortmannin	PI3K inhibitor
LY294002	PI3K inhibitor
SB202190	Cross-inhibition of the PI3K/mTOR and MAPKs pathway
SB203580	Inhibit autophagy by interfering with the trafficking of Atg9
MHY1485	mTOR activator

**Table 3 cancers-11-00533-t003:** Commonly used molecules to induce autophagy. Small molecules that have been identified as autophagy inducers and the primary mechanism of action.

Compound/Molecule	Autophagy Inducers
	Mechanism of Action
Rapamycin	mTORC1 inhibitor
Temsirolimus	mTORC1 inhibitor
Deforolimus	mTORC1 inhibitor
Everolimus	mTORC1 inhibitor
Metformin	AMPK activator
Obatoclax	Inhibitor of Bcl-2 family proteins
isoliensinine	Natural alkaloid
cepharanthine	Natural alkaloid
liensinine	Natural alkaloid
Perifosine	AKT inhibitor
Tat–Beclin-1 peptide	Releases beclin-1 into cytoplasm-regulate autophagosome formation
Lithium	Increase the levels of Beclin-1/VPS34 complexes
GDC-0980	Dual inhibitor of PI3K and mTORC1
GDC-0941	Inhibitor of class I PI3K
fluspirilene	Antagonists of L-type Ca2+ channels
verapamil	Antagonists of L-type Ca2+ channels
loperamide	Antagonists of L-type Ca2+ channels
nimodipine	Antagonists of L-type Ca2+ channels
amiodarone	Antagonists of L-type Ca2+ channels

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
