# Peer review of "The Interplay of Autophagy and Tumor Microenvironment in Colorectal Cancer—Ways of Enhancing Immunotherapy Action"

_cancers, 2019, doi:10.3390/cancers11040533_

Round 1

Reviewer 1 Report

The authors have made significant improvements to the manuscript but there are still errors throughout the text. For the figures, only partial changes were made, errors such as APC and MHC still persist in the latter half. The authors need to thoroughly correct these errors prior to publication.

Author Response

The authors have made significant improvements to the manuscript but there are still errors throughout the text. For the figures, only partial changes were made, errors such as APC and MHC still persist in the latter half. The authors need to thoroughly correct these errors prior to publication.

AUTHOR RESPONSE: We thank the reviewer. We have made the appropriate editing. A new version of Figure 1 was added. The impact of autophagy in exogenous and cross-presentation pathway of immunogenic peptides in antigen-presenting cell (APC) is portrayed correctly now. The mislabelled APC and MHC were corrected.

Reviewer 2 Report

In response to the authors,

The manuscript has been well improved, only minor comments are below.

1.Lines 32-33: A major mechanism which is strongly associated with tumorigenesis in different types of cancer, including CRC, is autophagy.

This sentence is too general. Angiogenesis, metastasis are also major mechanisms strongly associated with tumorigenesis…….Suggested correction is e.g.: Among others, autophagy is a major mechanism which is strongly associated with tumorigenesis in different types of cancer, including CRC.

2. Line 83: ’Cancer cells growth’ is not correct, e.g. In order to support the growth of cancer cells

3. Lines 81-84: In that context the inserted sentence is useless: KRAS is an importantan essential oncogene in CRC developing [242]. In the early stage of tumorigenesis autophagy has been characterized as a tumor suppressor mechanism. Cancer cells of KRAS- dependent tumors use autophagy in order to support cancer cells grown growth under the a stressful condition of in hypoxic regions of tumors [8]

Tumor suppressor role of autophagy should be emphasized separately, e.g.in the former paragraph it is already mentioned.

4. Line 101: This is not correct, a malignant cells is specialized or dedifferentiated: with specialized and dedifferentiated malignant cell types

5.Line 96: Reword this sentence: For years, it was a common belief, that previously solid cancers were considered the mass of homogenous cancer cells

6. Lines 263-270 can be misunderstood. Cross presentation should strengthen T-cell immunity. Presentation of extracellular antigens on MHC-I could support T-cell immunity. So, it is hard to understand what authors want to state.

7. Lines 278-291 are redundant (above is almost the same) and not clear about cross-presentation and autophagy. Use less and shorten sentences.

8. In the Figure 1, use 1A and B. in the figures (upright) APCs should be antigen presenting cells.

9. In the legend of Fig.1: ’in order to’ should not be separated

10. Line 371. chromosol….chromosomal

11. Grammatically is not correct this sentence: The immune-related objective response rate (ORR) and the 391 immune-related 6-month PFS rate were 40% and 78%, respectively, in the dMMR CRC patients, 0% 392 and 11% in the pMMR CRC patients.

12. Grammatically is not correct: In the update published on Lancet

13. Years of efforts lead….led

Author Response

The manuscript has been well improved, only minor comments are below.

1. Lines 32-33: A major mechanism which is strongly associated with tumorigenesis in different types of cancer, including CRC, is autophagy.

This sentence is too general. Angiogenesis, metastasis are also major mechanisms strongly associated with tumorigenesis…….Suggested correction is e.g.: Among others, autophagy is a major mechanism which is strongly associated with tumorigenesis in different types of cancer, including CRC.

AUTHOR RESPONSE: We thank the reviewer. We have made the appropriate editing. The changes in the manuscript are presented below:

Lines 34-36. “Among others, autophagy is a....of cancer, including CRC.”

2. Line 83: ’Cancer cells growth’ is not correct, e.g. In order to support the growth of cancer cells

AUTHOR RESPONSE: We thank the reviewer. We have made the appropriate editing. The changes in the manuscript are presented below:

Lines 77-79. “...in order to support the growth of cancer cells...”

3. Lines 81-84: In that context the inserted sentence is useless: KRAS is an important essential oncogene in CRC developing [242]. In the early stage of tumorigenesis autophagy has been characterized as a tumor suppressor mechanism. Cancer cells of KRAS- dependent tumors use autophagy in order to support cancer cells grown growth under the a stressful condition of in hypoxic regions of tumors [8]

Tumor suppressor role of autophagy should be emphasized separately, e.g.in the former paragraph it is already mentioned.

AUTHOR RESPONSE: We thank the reviewer. We have made the appropriate editing and the sentence has been incorporated in the previous paragraph. The changes in the manuscript are presented below:

Lines 77-79. “In addition, KRAS an essential....in hypoxic regions of tumors [8].”

4. Line 101: This is not correct, a malignant cells is specialized or dedifferentiated: with specialized and dedifferentiated malignant cell types

AUTHOR RESPONSE: We thank the reviewer. We have made the appropriate editing. The changes in the manuscript are presented below:

Lines 124. “...with specialized or dedifferentiated malignant cell types...”

5. Line 96: Reword this sentence: For years, it was a common belief, that previously solid cancers were considered the mass of homogenous cancer cells

AUTHOR RESPONSE: We thank the reviewer. We have made the appropriate editing and we rewrote the sentence. The changes in the manuscript are presented below:

Lines 118. “For years, solid cancers were considered as a mass of homogenous cancer cells [27].”

6. Lines 263-270 can be misunderstood. Cross presentation should strengthen T-cell immunity. Presentation of extracellular antigens on MHC-I could support T-cell immunity. So, it is hard to understand what authors want to state.

AUTHOR RESPONSE: We thank the reviewer for bringing this to our attention and offering us the opportunity to correct it. We rewrote the sentence (lines 359-362) and we highlight that autophagy is associated with cross-presentation pathway in DC.

7. Lines 278-291 are redundant (above is almost the same) and not clear about cross-presentation and autophagy. Use less and shorten sentences.

AUTHOR RESPONSE: We thank the reviewer. We have made the appropriate editing and we rewrote the sentence. The changes in the manuscript are presented below:

Lines 370-373. “Although, inhibition of autophagy up-regulates surface expression of MHC-I, it also changes the group of immunogenic peptides presented on MHC. Thus, the effect on surface expression of MHC-I and II is less well-confirmed, which has been best determined in the context of the so-called cross-presentation in DCs [93,100,101].”

8. In the Figure 1, use 1A and B. in the figures (upright) APCs should be antigen presenting cells.

AUTHOR RESPONSE: We thank the reviewer. We have made the appropriate editing. In the new figure we label exogenous pathways as 1A, cross-presentation as 1B and endogenous pathway as 1C. “APCs” is now corrected.

9. In the legend of Fig.1: ’in order to’ should not be separated

AUTHOR RESPONSE: We thank the reviewer. We have made the appropriate editing. The changes in the manuscript are presented below:

Lines 441. “...of proteins in order to APCs use them as...”

10. Line 371. chromosol….chromosomal

AUTHOR RESPONSE: We thank the reviewer. We have made the appropriate editing.

11. Grammatically is not correct this sentence: The immune-related objective response rate (ORR) and the 391 immune-related 6-month PFS rate were 40% and 78%, respectively, in the dMMR CRC patients, 0% 392 and 11% in the pMMR CRC patients.

AUTHOR RESPONSE: We thank the reviewer. We have made the appropriate editing and we rewrote the sentence. The changes in the manuscript are presented below:

Lines 579-582. “The immune-related objective response rate (ORR) and immune-related 6-month PFS progression-free surviva (PFS) rate were 40% and 78%, respectively, for mismatch repair–deficient (dMMR) colorectal cancers and 0% and 11% for mismatch repair–proficient colorectal cancers patients”

12. Grammatically is not correct: In the update published on Lancet

AUTHOR RESPONSE: We thank the reviewer. We have made the appropriate editing.

13. Years of efforts lead….led

AUTHOR RESPONSE: We thank the reviewer. We have made the appropriate editing and we rewrote the sentence

Reviewer 3 Report

The manuscript, though certainly improved, continues to have several technical and language-related issues. The issues are too numerous to discuss; hence a selected number of issues are discussed below. 

1.    Figure 1: In the top figure, the MHC II antigen presentation pathway is portrayed incorrectly. Cross-presentation is also incorrect. Please explain the two “circles” representing lysosomes. Please explain “Antigens Presented Cell” in the bottom figure.

2.  Line 116: The intension of this paragraph is to discuss T cells in tumor microenvironment. Do Th-1 cells activate CTLs in tumor microenvironment? Authors cite reference 346 to support this statement. Reference 346 does not exist in the manuscript. If authors are citing reference 34, how does this paper support the statement?

3.  Line 171-173: Authors are again requested to explain how the proliferation of T cells, which is mediated by CD3/CD28 signaling occurs in the tumor microenvironment in an antigen non-specific manner?

4. Line 260: Authors continue to refer to different factors without any context. For example, authors mention VPS34 without any context. Authors are requested to provide context to every protein/gene/factor mentioned in this review.

5.   Lines 263-266: Reference 90 shows that reduced autophagy leads to increased MHC I levels on the surface and enhanced CD8+ T cell response to viral antigens. The authors of this manuscript claim otherwise. Authors are requested to explain. 

6.     Line 116-118: Could authors cite some of the “many studies”?

7.     Line 129-130: How does reference 40 support this statement?    

Following minor points but they are indicative of substantial lack of attention:

8.  Line 131: “they are called tTreg” ???

9.  Line 143: Is the section number 2.34 or 2.3?

10.Lines 285-288: Please rewrite sentences in Lines 98-99, Lines 105-106, Lines 285-288. These sentences are difficult to understand.   

11. Line 356: How does this section fit in this review?

Author Response

The manuscript, though certainly improved, continues to have several technical and language-related issues. The issues are too numerous to discuss; hence a selected number of issues are discussed below.

1. Figure 1: In the top figure, the MHC II antigen presentation pathway is portrayed incorrectly. Cross-presentation is also incorrect. Please explain the two “circles” representing lysosomes. Please explain “Antigens Presented Cell” in the bottom figure.

AUTHOR RESPONSE: We thank the reviewer. We have made the appropriate editing.

i)                   A new version of Figure 1 was added. The impact of autophagy in exogenous and cross-presentation pathway of immunogenic peptides in antigen-presenting cell (APC) is portrayed correctly now. Moreover, autophagy in cooperation with endocytosis provides common nutrients and macromolecules for the use of the cell. They share common steps such as lysosomal degradation and the common terminal end-point. Autophagosomes fuses with early or late phase endosomes forming the “amphisomes”, before fusion into lysosomes, confirming the close relationship between these two pathways. The overlap linking endocytosis and autophagy can be also demonstrated by the localisation of endocytosed material contained by autophagic organelles [1,2,3].

1. C.A. Lamb, H.C. Dooley, S.A. Tooze, Endocytosis and autophagy: Shared machinery for degradation, BioEssays. 35 (2013) 34–45.

2. J.M.T. Hyttinen, M. Niittykoski, A. Salminen, K. Kaarniranta, Maturation of autophagosomes and endosomes: A key role for Rab7, Biochimica et Biophysica Acta - Molecular Cell Research. 1833 (2013) 503–510.

3. D.J. Klionsky, Autophagy: from phenomenology to molecular understanding in less than a decade., Nature Reviews Molecular Cell Biology. 8 (2007) 931–937.

ii)                 The two circles represented the fusion of autophagophore with the lysosome. We change the graph and the autophagasome is represented now as a circle with different color.

iii)               We changed “the Antigens Presented Cell” to “Antigen-presenting Cell”

2.  Line 116: The intension of this paragraph is to discuss T cells in tumor microenvironment. Do Th-1 cells activate CTLs in tumor microenvironment? Authors cite reference 346 to support this statement. Reference 346 does not exist in the manuscript. If authors are citing reference 34, how does this paper support the statement?

AUTHOR RESPONSE: We thank the reviewer for bringing this to our attention and offering us the opportunity to correct it. We rewrote the sentence in order to better explain our statement. Reference 34 mentions this point in page 10, last paragraph of discussion. The changes in the manuscript are presented below:

Lines 138-140. “The abundance of T-cells is a positive prognostic factor for the response of chemotherapy and immunotherapy in cancer patients especially at early stage of the disease, where the patients have a strong effector T cell response and more frequently present a high Immunoscore. [33,34].

3.  Line 171-173: Authors are again requested to explain how the proliferation of T cells, which is mediated by CD3/CD28 signaling, occurs in the tumor microenvironment in an antigen non-specific manner?

AUTHOR RESPONSE: We thank the reviewer for bringing this to our attention and offering us the opportunity to correct it. The changes in the manuscript are presented below:

Lines 229-232. “In tumor microenvironment, MDSCs are converted into nonspecific suppressor cells through up-regulating of iNOS and arginase I. These enzymes are known to be actively involved in T cell suppression in a way that do not require antigen-specific contact between MDSC and T cells to inhibit their function [62].”

4. Line 260: Authors continue to refer to different factors without any context. For example, authors mention VPS34 without any context. Authors are requested to provide context to every protein/gene/factor mentioned in this review.

AUTHOR RESPONSE: We thank the reviewer for bringing this to our attention and offering us the opportunity to correct it. VPS34 is the vacuolar protein sorting-associated protein 34, an essential component for autophagy initiation. The changes in the manuscript are presented below:

Lines 357-358. “Equally, DCs from mice lacking VPS34 (vacuolar protein sorting-associated protein 34) expressed...”

5.   Lines 263-266: Reference 90 shows that reduced autophagy leads to increased MHC I levels on the surface and enhanced CD8+ T cell response to viral antigens. The authors of this manuscript claim otherwise. Authors are requested to explain.

AUTHOR RESPONSE: We thank the reviewer for bringing this to our attention and offering us the opportunity to correct it. We rewrote the sentence. The changes in the manuscript are presented below:

Lines 355-356. “This adjustment was attributed to the slower internalization of classical MHC class I molecules leads to increased CD8+ T cell stimulation [90].”

6.     Line 116-118: Could authors cite some of the “many studies”?

AUTHOR RESPONSE: We thank the reviewer for bringing this to our attention. We added three more references in order to support the statement.

37.  Hu, Z.; Ma, Y.; Shang, Z.; Hu, S.; Liang, K.; Liang, W.; Xing, X.; Wang, Y.; Du, X. Improving immunotherapy for colorectal cancer using dendritic cells combined with anti-programmed death-ligand in vitro. Oncol Lett. 2018, 15, 5345-5351.

38.  Pauken, K.E.; Wherry, E.J. Overcoming T cell exhaustion in infection and cancer, Trends Immunol. 2015, 36, 265–276. 

39.  Singh, P.P.; Sharma, P.K.; Krishnan, G.; Lockhart, A.C. Immune checkpoints and immunotherapy for colorectal cancer. Gastroenterol Rep (Oxf). 2015, 3, 289-297.

7.     Line 129-130: How does reference 40 support this statement?  

AUTHOR RESPONSE: We thank the reviewer for bringing this to our attention and offering us the opportunity to better explain our statement. We rewrote the sentence. The changes in the manuscript are presented below:

Lines 152-157. “The role of Tregs (regulatory T cells) is controversial because of the genetic and phenotypic differentiation of T-cells. The Treg-specific DNA hypo-methylated regions contribute to the stable expression of Treg function-associated key genes including Foxp3. Accordingly, FoxP3 strongly represses different genes including Il2, contributing to Treg suppressive activity. In tumor, it is critical to deplete FOXP3high CD45RA_ CD25high effector Treg cells, which are firmly installed with the Treg-type hypo-methylation and most suppressive [43].”

Following minor points but they are indicative of substantial lack of attention:

8.  Line 131: “they are called tTreg” ???

AUTHOR RESPONSE: We thank the reviewer. We have made the appropriate editing. The changes in the manuscript are presented below:

Lines 157-158. “The origin of Tregs can be either directly from the thymus (tTreg) or by peripheral differentiation (pTreg) of conventional T lymphocytes [44].”

9.  Line 143: Is the section number 2.34 or 2.3?

AUTHOR RESPONSE: We thank the reviewer. The section number in line 130 is 2.2 and in line 162 is 2.3.

10. Lines 285-288: Please rewrite sentences in Lines 98-99, Lines 105-106, Lines 285-288. These sentences are difficult to understand. 

AUTHOR RESPONSE: We thank the reviewer. We have made the appropriate editing and we rewrote the sentences. The changes in the manuscript are presented below:

Lines 118-123. “Cancer evolution and resistance to...mutations background driving their growth [28].”

Liens 129-132. “It is widely accepted that tumorigenesis is....mesenchymal cells, fibroblasts and immunogenic cells [30,31].”

Lines 435-438. “Inhibition of autophagy modified the presentation of the....antigen-presenting cells (APCs) is presented in Figure 1.”

11. Line 356: How does this section fit in this review?

AUTHOR RESPONSE: We thank the reviewer for bringing this to our attention and offering us the opportunity to better explain our statement. We incorporated the sentence in the previous paragraph in order to introduce a new topic about how microsatellite status affects immunotherapy.

Lines 565-568. “It is well known that MSI phenotype in CRC varies according to the stage of the disease. CRC patients……rate for CRC patients with mismatch repair deficiency is less than 50 % [125] and less than 30% for gastroesophageal cancer [125,126].”

This manuscript is a resubmission of an earlier submission. The following is a list of the peer review reports and author responses from that submission.

Round 1

Reviewer 1 Report

This manuscript has very significant shortcomings. 

Major issues:

1.     Autophagy and tumor microenvironment are the most significant keywords of this article. However, this manuscript doesn’t describe any details of the process of autophagy. Authors mention several factors involved in autophagy throughout the review without providing any background or context. The section on tumor microenvironment and the cells found within it needs elaboration. 

2.    Review articles must also introduce readers to original research articles that are significantly to the field. This review manuscript cites too many previously published review articles instead. 

3.    The manuscript contains significant language-related issues.

4.    The numerous minor issues (below) are collectively a major issue.

Minor issues

1.     Line 36-37: How does reference (reference 4) support the statement?

2.    Line 53: How does reference 17 support the statement that begins this line?

3.    Lines 49-50: The authors state that increased levels of autophagy are linked with poor patient survival rates. However, in the next line authors state the opposite. Authors should elaborate both the points to provide clarity.

4.    Line 75: “major’s” should be changed to “major”.

5.    Lines 76-78: The two sentences are incomprehensible.

6.    Lines 84-85: The sentence says that tumorigenesis is a result of a disturbance in the balance between the cells within the tumor microenvironment. Please explain.  

7.     Line 107: Define normal T lymphocytes

8.    Lines 117 and 188: Define the abbreviations.

9.    Lines 139-140: What do the authors mean by CD3/CD28 T cell proliferation?

10. Line 232: It seems like the authors want to convey that cross-presentation capability of DCs is affected by the level of autophagy within cells. Please specify which cells: the DC or other antigen donor cells?   

11.  Line 243: What is cross-presence?

12.  Line 243: Cross-presentation is defined, again. The point and the words/sentence is redundant; the exact sentence appears earlier in lines 230-231. 

13.  Line 246: What are non-crossover DCs?

14.  Line 245 (the sentence that begins this line): The is redundant (same point and similar words as in line 232).

15.  In Figure 1, the top label says ANTIGEN PRESENTED CELL (APC). Perhaps, the authors mean ANTIGEN PRESENTING CELLS (APC). In the same figure, the antigens presented on MHC I appear to be membrane proteins of the same cell. Is this the intention? If it is, the figure does not agree with the preceding text. The text repeatedly mentions that autophagy affects cross-presentation. However, the figure depicts presentation of self-antigens and not cross-presentation of foreign, extra-cellular antigens. Change the figure to show cross-presentation. Also change the figure to include MHC II presentation. Describe the role of autophagy in presentation of self-antigens in context of MHC I and MHC II in the text to bring accord to text and figure.

16.    Line 292 (new sentence beginning on this line): Should the word deplete be replaced with depletion? In the same line, explain smooth operation of Tregs.       

17.    Line 310: Define mCRC.

18.    Line 326-328: How was the antibody identified in patients? The sentence needs to be re-written to express what the authors want to convey. 

19.    Line 389: Authors should discuss new possibilities instead of simply pointing the readers to yet another review on the same subject.

20.   Lines 406-408: The two sentences contradict each other. The first sentence is incorrect.

21.    Line 405: Until this point, the review suggests that autophagy plays a role in resistance of tumors to different therapies (lines 367-368). Section 5.2 discusses activators of autophagy as anti-tumor therapeutic agents. The authors should include a paragraph at the beginning of this section to introduce this idea that seems counterintuitive in context of previous sections of this review. Specifically, authors should discuss how activation of autophagy may improve outcomes of other therapies.

22.   Lines 415-417: Use an activator and an inhibitor simultaneously? Authors should elaborate.

Reviewer 2 Report

Comments to the authors

The interplay of autophagy and tumor microenvironment - Ways of enhancing immunotherapy action, Cancers_472090

The manuscript from Koustas et al. dissects the relation of autophagy and anti-cancer immunotherapy. The authors cover the following topics: 1. Introduction, 2. The major's players in the tumor microenvironment, 2.1. The heterogeneity of tumor microenvironment, 2.2. The role of T-lymphocytes, 2.3. Tregs - a specialized subpopulation of T cells, 2.4. The role of Tumor-Associated Neutrophils (TANs), 2.5. Cancer-associated fibroblasts (CAFs), 2.6. Angiogenesis and neo-vascularization process in tumor stroma, 2.7. Other immune cell types in tumor microenvironment, 3. The role of autophagy in stroma developing, Inflammation and Immunity response, 3.1. The critical role of autophagy in inflammation, 3.2. Hypoxia-induced autophagy in the tumor microenvironment, 3.3. The cross-talk between autophagy and antigen presenting cells, 3.4. Autophagy - a key regulator of T-cell activation, 4. The current state of immunotherapy in CRC patients, 5. Targeting autophagy - A promising anti-cancer strategy, 5.1. The main autophagy inhibitors in cancer therapy, 5.2. Activators of Autophagy for Cancer Therapy, 6. Conclusions

The authors focus on a scientifically important field which connects autophagy and anti-tumor immunity. Due to the high mortality rate of solid malignancies in adults both the dissection of autophagy and immunotherapy fits in the scope of Cancers journal.

The different subsections and number of references give a general overview about autophagy and anti-tumor immunity. Structure of the paper is logic and well organized. Anyhow, there are too serious grammatical errors, controversial sentences.

In my overall opinion the paper can not be accepted in the present form, but it’s worth to improve it. I would highly ask the authors for the improvement of the English grammar and style (consult with a native English scientist) before resubmission. Please, pay attention for a thorough revision in terms of English grammar and using scientific terminology.

I have the following suggestions to improve the manuscript: 

Comments:

1.      The authors should emphasize better the focus on colorectal cancer (CRC) in the abstract and title also.

2.       Raws 31-32: This sentence is too general, what is the link between different types of cancers and autophagy? Autophagy plays a role among other mechanisms, this should be emphasized. A major mechanism which is strongly associated with tumorigenesis in different types of cancer, including CRC, is autophagy.’

3.      Raws 34-36: Authors should rewrite. e.g. ’Numerous steps related to autophagy…’ or different instead of: The numerous autophagy steps….

4.      Raw 37: use ’is a fundamenta’l instead of ’is the fundamental’

5.      Raw 41: use ’genes have been reported with high expression’ or differentinstead of ’genes are appeared upregulated’

6.      Raw 43: There is no comma after many: ’The last years, many studies support….’

7.      Raw 45: Mention more mechanisms: ’ such as proliferation, angiogenesis, metastasis etc…Cite accordingly.

8.      Autophagy can be tumor promoter and even can be endowed with antitumor activity also. Please, dissect these controversial data clearly between Raws 51-74.

9.      Raws 76-77. There are several papers about intratumor heterogeneity of cancer cells. (e.g. Gerlinger, Intratumor Heterogeneity and Branched Evolution Revealed by Multiregion Sequencing, NEJM 2012). You can not simply write homogenous in that context. You may intended to write that previously solid cancers were considerd the mass of homogenous cancer cells, that would be right.

10.  Raw 79: Malignant cells can be specialized and dediferentiated also, not exclusively specialized.

11.  Raw 83: Write this: …..of solid tumors are composed of not only by malignant cells but also by fibroblasts…..

12.  Raw 94: May be grading would be more appropriate instead of ’classification’

13.  Two sentences are too overlapping: raws93-94 and 96-97. Avoid to repeat the same meaning within a paragraph.

14.  Prepare a lis for abbreviations: MMRd, MSI-H, MSS etc. throughout the paper

15.  2.3. Tregs should be within 2.2 and discussed in one subchapter

16.  Raw 113: ’maintain peripheral tolerance’ instead of keep the balance’

17.  Raw 114:’….are crucial in the inhibition…’

18.  The subchapter of 2.4 is very important, this title should be: ’ The role of tumor associated myeloid cells’

19.  One short sentence and citation (e.g. Szebeni et al Pro-Tumoral Inflammatory Myeloid Cells as Emerging Therapeutic Targets, IJMS 2016) is missing about the targeting of TAMs and MDSCs in the tumor microenvironment in Raw 121.

20.  Raw 122-123: There is no verb: The anti-tumorigenic…..

21.  Raw 147: The initiation of autophagy increases

22.  Raw 150: Write this or different: CAFs represent a heterogenous group of cells. They are responsible for the remodeling of the ECM and support the invasion……

23.  Raw 152-153……have been already used instead of have already used

24.  Raw 154: presence of CAFs with poor…..

25.  Please correct these type of grammatical errors throughout the paper, I can not list all and suggest corrections.

26.  Raw 173: Do you mean hydrostatic pressure or stress conditions?

27.  Section 3.1 should cover the relation of autophagy and inflammation but is more about colorectal cancer. Change the title or this content. Cite a seminal paper by Fukata, Toll-like receptor-4 (TLR4) promotes the development of colitis associated colorectal tumors, Gastroenterology

28.  Raw 216: Activation of the anticancer T-cells

29.  What is 3-MA, prepare a list of abbreviations, please.

30.  Raws 226-232. This is not logic. Cross-presentation should support the activation of T-cells (MHC-I-CD4, MHC-II-CD8)

31.  Figure 1: APCs are antigen presenting cells

32.  Raws 406-408: These are controversial sentences, clarify please that these are autophagy inhibitors (rapamycin, rapalog).

Reviewer 3 Report

Koustas et al. present a description of studies on autophagy and its role in the tumor microenvironment as well as immunotherapy. The review is poorly structured, does not have a logical flow, contains spelling and grammatical errors in the text and figures, have repetitive segments, making it difficult for the reader to extract any useful information. It will take a significant amount of effort to improve the manuscript to a level that will be useful for its audience. Some more specific comments include:

1)     The title implies a broad review that is applicable to all cancers, but there is a narrow emphasis on CRCs. While “ways of enhancing immunotherapy action” is part of the title, it is difficult to extract any explicit information on this.

2)     Abbreviations are not described in the first instance (Lines 99, 313).

3)     Line 107, thymus and peripheral Tregs were both abbreviated as tTreg.

4)     Lines 226-237 are almost redundant with lines 238-253. Similar instances of such repetition should be removed.

5)     MHCI mislabeled as MCHI in figure 1. APC-definition needs to be corrected.

6)     In Table 2, a more extensive list of autophagy inhibitors should be included. (e.g. Spautin-1, PMID: 21962518)

7)     Lines 415-419, studies showing improved therapeutic efficacy upon metformin treatment and autophagy inhibition should also be discussed (PMID: 29938573).

8)     In section 5, it would also be useful to include discussion on a study showing that anti-tumor adaptive immunity remains intact following autophagy inhibition (PMID: 27775547).